# Temperature alters the predator-prey size relationships and size-selectivity of Southern Ocean fish

Patrick Eskuche-Keith [1,2] ✉, Simeon L. Hill [2], Lucía López-López [3], Benjamin Rosenbaum [4,5], Ryan A. Saunders[2], Geraint A. Tarling [2] & Eoin J. O'Gorman [1]

A primary response of many marine ectotherms to warming is a reduction in body size, to lower the metabolic costs associated with higher temperatures. The impact of such changes on ecosystem dynamics and stability will depend on the resulting changes to community size-structure, but few studies have investigated how temperature affects the relative size of predators and their prey in natural systems. We utilise >3700 prey size measurements from ten Southern Ocean lanternfish species sampled across >10° of latitude to investigate how temperature influences predator-prey size relationships and size-selective feeding. As temperature increased, we show that predators became closer in size to their prey, which was primarily associated with a decline in predator size and an increase in the relative abundance of intermediate-sized prey. The potential implications of these changes include reduced top-down control of prey populations and a reduction in the diversity of predator-prey interactions. Both of these factors could reduce the stability of community dynamics and ecosystem resistance to perturbations under ocean warming.

Global warming represents a major threat to the structure and functioning of ecosystems. One possible consequence of rising temperatures is a decrease in body size across many species and communities[1]. At the individual level, warming alters the physiology of organisms and is likely to reduce body sizes within populations as organisms attempt to maintain metabolic functioning[1,2]. At the community level, warming may alter assembly processes through environmental filtering, competition, or trophic interactions, which may result in communities dominated by smaller-bodied species[1,3]. The subsequent impacts on population abundances and species interactions can drive changes to structure and function at the ecosystem scale[4]. Aquatic ectotherms such as fish are particularly susceptible to temperature-induced reductions in body size, due to the lower rates of oxygen diffusion in water and the energetic costs associated with maintaining water flow over surfaces[5]. Additionally, gape-limited feeding means that many fish

species display ontogenetic changes in prey selection, with larger predators consuming larger, more energetically valuable prey[6,7]. Declines in prey size with warming may therefore reduce the rates of energy acquisition by larger predators, resulting in reduced fish growth and smaller overall body sizes within populations[8]. Furthermore, such altered prey size distributions may favour smaller-sized predator species, providing them with a competitive advantage and thereby shifting the fish community composition towards smaller body sizes[9]. Evidence from the last interglacial period suggests that fish communities experienced declining body size in response to warmer conditions[10,11], and the average size of contemporary fish is expected to show a similar pattern under the current rate of global warming[12]. However, there is currently little understanding of how these changes will impact the structure and stability of marine ecosystems.

[1]School of Life Sciences, University of Essex, Colchester, UK. [2]British Antarctic Survey, Cambridge, UK. [3]Ecosystem Oceanography Group (GRECO), Oceanographic Centre of Santander (CN IEO, CSIC), Santander, Spain. [4]EcoNetLab, German Centre for Integrative Biodiversity Research (iDiv) Halle-Jena-Leipzig, Leipzig, Germany. [5]Institute of Biodiversity, Friedrich Schiller University Jena, Jena, Germany. ✉ e-mail: patrickaekeith@gmail.com

Body mass is a key life-history trait that determines factors such as consumption rates, handling times and gape size[13,14]. As such, body mass provides an important link between individual physiology and food web structure and is therefore often used to parameterise models of population dynamics and energy flow within ecosystems[15,16]. In the marine environment, predators are generally larger than their prey, and the predator-prey mass ratio (PPMR) is a good predictor of trophic interactions. For example, allometric diet breadth models accurately predict who eats who in aquatic ecosystems[13], whilst declines in PPMR typically result in lower per capita interaction strengths as predators are able to gain the same amount of energy by consuming fewer large prey[17]. At the community level, larger ectotherms may decline in size more rapidly than smaller ectotherms with warming as a result of their reduced surface area to body mass ratio and the associated challenge of maintaining a higher metabolic rate[5,18]. This is particularly true for the marine environment, where larger fish and invertebrates display the strongest temperature-size responses[19]. If warming causes a greater decline in the size of ectotherm predators relative to that of their smaller prey (i.e. changes in community size structure), the average PPMR might decrease, with consequences for interaction strengths and thus energy flow through marine ecosystems.

The physiological basis for temperature effects on PPMR at the community level may be complicated by behavioural responses to environmental change. For example, predators may select for more nutritious (larger) prey in an effort to increase per capita energy intake under energetically stressful conditions, thus reducing their PPMR[20,21]. Alternatively, predators might feed in a more density-dependent manner, consuming a greater proportion of abundant but relatively smaller prey and thereby increasing PPMR. Importantly, behavioural responses are unlikely to be uniform across predator body sizes, given the different dietary niches of small and large organisms and their differential susceptibility to warming. Previous research has identified

variable size-dependent relationships between PPMR and temperature, such that both systematic increases[22] and decreases[23] to per capita interaction strength are possible.

It is clear we still have limited understanding of how temperature-driven changes in body size may alter community-level feeding relationships, and it is vital to address this knowledge gap if we are to predict ecosystem responses to warming. This is particularly true for the Southern Ocean, which is experiencing widespread environmental changes including rapid regional warming in areas such as the western Antarctic Peninsula[24] and northern Scotia Sea[25]. The Southern Ocean supports a diverse array of higher predator populations including seabirds, seals, penguins and whales, with a food web largely centred around krill (particularly *Euphausia superba*)[26]. However, it is expected that krill will shift their distribution southward in response to ocean warming[27], with potentially drastic consequences for many regional predator populations unless other suitable prey are available[28]. Previous research has identified mesopelagic lanternfish (Family Myctophidae, hereafter myctophids) as one such potential alternative resource, due to their extremely high biomass and their role in supporting energy flow to higher predators including seals and penguins during periods of low krill availability[29]. Additionally, myctophids themselves are major generalist consumers of prey including krill, amphipods and copepods, and therefore exert significant influence over food web dynamics[30]. Myctophids are strongly size distributed in the Southern Ocean, with smaller species and individuals found at lower (warmer) latitudes[31], and they display clear size-selectivity in their feeding[32,33]. Warming may therefore alter the size distribution of myctophids and the size relationships between these predators and their prey, and it is important that we understand what these likely changes will be in order to model ecosystem responses.

In this study, we assessed the relationship between temperature and the relative sizes of myctophids and their prey using a dataset of 1576 stomachs and 3707 prey size measurements from 10 myctophid species sampled across >10° of latitude in the Southern Ocean (Fig. 1). We hypothesised that myctophids would exhibit a decline in PPMR with increasing temperature, due to (1) a greater decrease in the size of these predators versus their prey, and/or (2) predators selecting for larger prey as temperature increases.

## Results and discussion

PPMR declined by ~11% per °C increase in sea surface temperature (SST), associated with a significant decline in predator body size at a rate of ~6% per °C and no coherent trend in the mean body size of prey in the diet (Fig. 2 and Table 1). Chlorophyll *a* was initially used as a further explanatory variable but it was not significant in any model and was therefore excluded during model selection (Tables S1–S7). The same general results were found when temperature at ~1000 m (the estimated maximum of myctophid depth distributions) was considered instead of SST (Fig. S1 and Tables S8–S15). A similar decline in predator body size was also found when using a larger dataset of fish body masses (*n* = 6143, the majority without stomach content data; Fig. S2 and Tables S16–S18). In all, 7 of the 10 myctophid species also displayed significant declines in size with increasing temperature (Fig. S3 and Tables S19–21). Together, these results suggest that the decline in PPMR is associated with a greater decrease in the size of these predators relative to their prey as temperature increases.

The effect of declining PPMR on interaction strengths will depend on the interactive effects of temperature and body mass on metabolism and consumption[34], making it difficult to predict the consequences for ecosystem stability. It has previously been found that temperature alters the directionality and shape of the relationship between PPMR and predator attack rate and prey handling time, with low PPMR destabilising community dynamics under warming due to elevated predation rates at low prey density[34]. Additionally, when declines in body mass under warming are restricted to isolated trophic

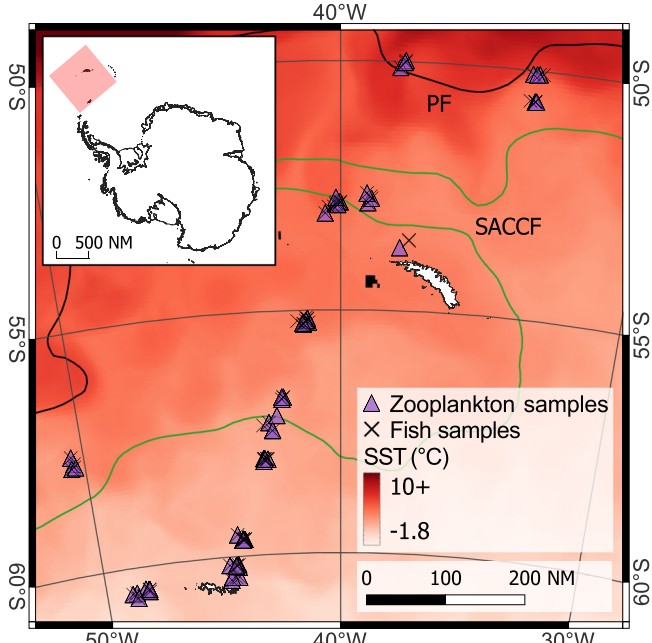

**Fig. 1 | Map of the study region displaying the locations of myctophid (black crosses) and zooplankton (purple triangles) sampling stations.** The interannual average position of key oceanic fronts are also displayed (PF = Polar front; SACCF = Southern Antarctic Circumpolar Current front). Temperature data represents the mean value from 15th March – 15th April 2009 from the Copernicus Global Ocean Physics Reanalysis (GLORYS12)[60]. Map projection is WGS84/Antarctic Polar Stereographic. Black fill represents missing temperature data. Map produced using QGIS 3.28 Firenze.

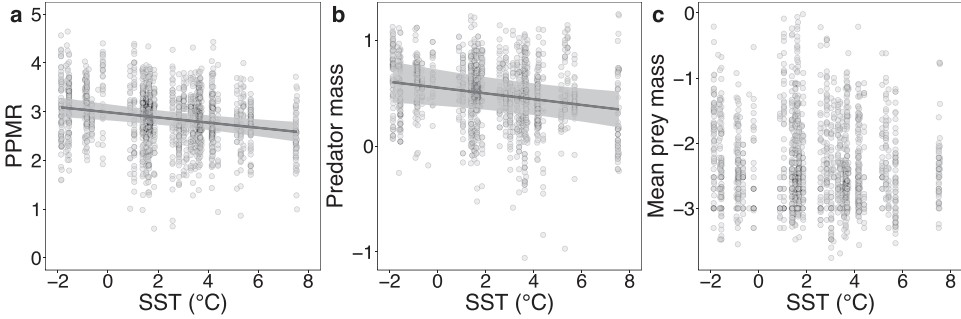

**Fig. 2 | Effects of temperature on predator and prey body mass. a** partial residual plot from a linear mixed model of the effect of sea-surface temperature (SST) on prey-averaged predator-prey mass ratio (PPMR); (**b**) partial residual plot from a linear mixed model of the effect of SST on predator body mass; (**c**) scatterplot of the relationship between SST and abundance-weighted average prey mass in predator stomachs. $Y$-axis values are in $\log_{10}$ g. Lines represent predicted values at each SST. Shading represents 95% confidence intervals. Source data are provided as a Source Data file.

levels, community stability is expected to be reduced[35], possibly due to lower top-down control of prey populations[36]. However, while the reduced ingestion efficiencies and higher metabolic costs associated with higher temperatures are expected to make predator populations increasingly vulnerable to starvation, this effect is exacerbated under high PPMRs[37], therefore the observed decline in predator size with warming may in fact provide a buffer against population crashes. Ultimately, the effects of warming and PPMR on the strength of interactions will depend on factors including predator and prey identity, predator body size and thermal tolerance. Further investigations of the combined effects of temperature and PPMR on interaction strengths will be important for determining the possible consequences of altered size-structuring of predator-prey interactions for the stability of ecological communities. This could be facilitated through the application of ecosystem flux or dynamical population models[38,39].

The observed decline in predator size with increasing temperature fits the wider expectation that a primary response of ectotherm vertebrates, including Southern Ocean myctophids, to warming should involve a reduction in individual body size and shifts in overall community size structure[2,5]. Declines in size at the individual level are thought to facilitate continued persistence with warming by minimising the extent to which metabolic rate must increase to match the greater energetic demands of the environment[40]. Changes in community size structure may also be the result of a combination of physiological and competitive processes which result in species of a certain size range becoming dominant[9]. There was a significant increase in Shannon diversity of the myctophid community with increasing SST, associated with a shift in species abundances from communities dominated by a few large-bodied species (e.g. *Electrona antarctica*) at cold high latitudes to a more even distribution of abundances in the more northerly warmer regions (Fig. S4 and Tables S22–24), as previously documented[41,42]. This indicates that the link between temperature and body size at the community level may be driven in part by community assembly processes that select for species of different sizes as it becomes warmer, e.g., smaller predators are able to outcompete larger ones under the altered prey size distribution and relatively lower metabolic demands. However, our analyses of the relationship between body mass and SST at the population level also revealed significant declines in size with increasing temperature for many of the myctophid species, both for large-bodied taxa such as *E. antarctica* and for small species like *Krefftichthys. anderssoni* (Fig. S3 and Tables S19–21). The observed trends at the community level therefore are not explained by community assembly processes alone, but also by temperature effects on populations, likely mediated by physiological responses to warming.

Under both moderate and high emissions scenarios, Antarctic waters are expected to become increasingly favourable for smaller, sub-Antarctic myctophid species, likely altering community diversity and size structure[43]. Such changes may reduce their suitability as prey for predators such as penguins and seals, with knock-on effects on these higher predator populations and food web dynamics[44]. Additionally, many myctophid species display size-selective feeding, with a switch from euphausiids and fish to smaller copepods as their body size decreases[33]. Thus, a reduction in the average size of myctophids may alter the diversity and size distribution of the prey community as predation rates on different species change[45,46]. Furthermore, smaller species are generally expected to have fewer feeding interactions across a more restricted range of trophic levels, which could alter the distribution of energy flow by reducing network complexity and trophic redundancy[47].

To investigate the evidence for size-selective feeding behaviour that could further underlie the decline in PPMR with temperature, we conducted an analysis of dietary size preferences for prey in the environment in relation to predator body size class and temperature (see Methods). Predator size and SST had a significant interactive effect on preferred prey size, with small predators feeding on relatively larger prey and large predators feeding on relatively smaller prey in warmer regions (Fig. 3a and Table 2). This partially supports hypothesis 2, that predators will select for larger prey in warmer environments, but not for the largest fish. This result may be explained by an increase in the relative abundance of intermediate prey sizes within the range of body masses commonly consumed by the fish (Fig. 3b).

## Table 1 | Model statistics for the effect of temperature on predator and prey body masses

| Model | Coefficient | Estimate | SE | DF | t-value | p-value |
|---|---|---|---|---|---|---|
| PPMR | Intercept | 2.988 | 0.085 | 1550 | 35.251 | <0.0001 |
| | SST | −0.053 | 0.015 | 1550 | −3.601 | 0.0003 |
| $R^2m = 0.049$, $R^2c = 0.493$ | | | | | | |
| Predator body mass | Intercept | 0.552 | 0.087 | 1550 | 6.346 | <0.0001 |
| | SST | −0.027 | 0.011 | 1550 | −2.482 | 0.0132 |
| $R^2m = 0.024$, $R^2c = 0.978$ | | | | | | |
| Mean prey body mass | Intercept | −2.371 | 0.069 | 1551 | −34.249 | <0.0001 |
| $R^2m < 0.001$, $R^2c = 0.455$ | | | | | | |

Output from linear mixed models with predator-prey mass ratio (PPMR), predator body mass and abundance-weighted average prey body mass in predator stomachs as response variables (all $\log_{10}$). SST represents sea-surface temperature. $R^2m$ and $R^2c$ represent the Nakagawa's marginal and conditional model $R^2$ values, respectively.

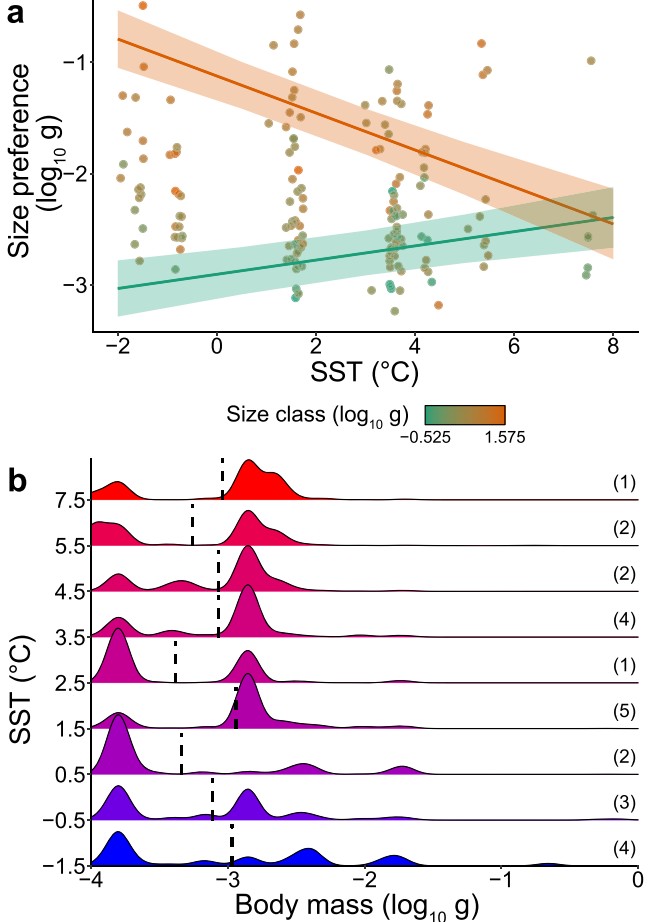

**Fig. 3 | Temperature effects on predatory size preferences and body mass distribution of prey in the environment. a** Predicted interactive effect of SST and fish size class on myctophid average preferred prey size. Lines represent predicted values at each SST, for the largest and smallest predator size classes. Shading represents 95% confidence intervals. Points are coloured according to size class, jittered slightly for clarity. **b** Density plots of zooplankton body mass distribution in the environment within size range commonly consumed by the myctophids, grouped into 1 °C temperature bins. Dashed lines represent abundance-weighted average body mass. Y-axis indicates central temperature value for each bin. Values in brackets indicate number of hauls. Note: in panel b, large prey sizes (above approx. $-2 \log_{10} g$) are present at all temperatures but extremely low abundance relative to smaller organisms prevents them from being visible. Source data are provided as a Source Data file.

**Table 2 | Model statistics for the effect of temperature on predatory size preferences**

| Coefficient | Estimate | SE | DF | t-value | p value |
|---|---|---|---|---|---|
| Intercept | −2.460 | 0.194 | 139 | −12.673 | <0.0001 |
| SST | 0.006 | 0.021 | 139 | 0.295 | 0.7681 |
| Size class | 0.848 | 0.109 | 139 | 7.795 | <0.0001 |
| SST*Size class | −0.110 | 0.031 | 139 | −3.483 | 0.00071 |
| $R^2m = 0.298$, $R^2c = 0.801$ | | | | | |

Output from a linear mixed effects model with mean preferred prey size ($\log_{10}$) as the response variable and sea-surface temperature (SST) and predator size class ($\log_{10}$) as explanatory variables. $R^2m$ and $R^2c$ represent the marginal and conditional model $R^2$ values, respectively.

meeting their higher energetic demands under warmer conditions. These changes in size-selectivity may also explain the increasing prevalence of smaller myctophid species in warmer regions (Figure S4), as they can capitalise on the available prey field and outcompete their larger counterparts. The increasing dominance of smaller myctophids, which feed preferentially on larger prey in warmer regions, is likely to drive the observed decline in overall PPMR across the predator community. Thus, we suggest that the observed patterns in myctophid size and foraging with temperature are likely to be the result of a combination of interacting processes acting at both the population and community levels, and we encourage further efforts to disentangle them. Overall, our results highlight the importance of considering the size structuring of biotic interactions and plasticity of size-based foraging behaviour when investigating the possible consequences of environmental change for community structure and composition.

We investigated a temperature gradient across a large spatial scale (>10° of latitude) rather than directly testing the effects of temperature change over time. Such temporal changes are difficult to investigate in-situ, but mesocosm experiments could provide insight into how rapid warming affects species body sizes and biotic interactions. However, the results of such studies primarily relate to the plastic responses of individuals over the short-term, which may differ from the adaptive responses of populations to sustained gradual warming over the multi-decadal timescales that are relevant to ongoing climate change. In contrast, given the historically stable temperatures of the Southern Ocean[49], our space-for-time substitution represents the long-term eco-evolutionary adaptation of predator and prey communities. One potential caveat of our approach was the use of sea-surface temperatures to represent the environmental conditions experienced by the myctophids, as temperatures at depth may differ from those at the surface. Indeed, while a positive relationship between latitude and temperature is still apparent at approximately 1000 m depth, the trend is weaker than at the surface (Fig. S2). When substituting SST with the temperature at depth in our analyses, however, the results are consistent (Figure S1, Tables S8-S15), suggesting that the observed relationships hold across the depth range that myctophids are thought to inhabit.

As our oceans continue to warm, significant changes to the size structuring of marine communities are likely to occur in many regions, and the use of dietary preference analyses such as this will be useful for disentangling the interactive effects of behaviour and physiology on the feeding ecology of key species and functional groups. Myctophids are one of the most abundant fish families globally and a major component of many pelagic food webs, from the poles to the tropics[49,50]. The insights gained in this study therefore have relevance for other open ocean systems, including those near the equator where warming is expected to drive strong declines in body size and changes to the distribution of many mesopelagic species[50,51]. Changes in species composition with temperature may also alter community PPMR in unexpected ways, as it has previously been found that the relationship between individual body mass and PPMR varies between taxa, due to factors such as morphology and feeding strategy[52]. It will therefore be

Our results suggest that temperature influences the size-structuring of feeding relationships within the Southern Ocean mid-trophic community through a combination of density-dependence and active selection. Under colder conditions, large predators appear to select for relatively abundant, large, energetically valuable prey while small predators feed on small prey. Under warmer conditions, the shift in the distribution of suitable prey sizes towards intermediate body masses restricts the feeding behaviour of large predators and forces them to feed sub-optimally on smaller prey while small predators actively select for these abundant intermediate prey sizes, possibly because they provide greater per capita energy intake. This reduction in prey size diversity could constrain the foraging niches of smaller and larger predators, increasing competition and, under the general expectation that food web complexity promotes predator population stability[48], potentially destabilising predator-prey dynamics. Larger predators may also be forced to feed on prey that are smaller than their optimal foraging niche, thus preventing them from

important to expand these analyses to other regions and taxa to provide an overview of the generality of the observed relationships.

Rising metabolic costs and oxygen limitation resulting from ocean warming are expected to drive declines in the body size distribution of many marine ectotherms[2,5], and we sought here to investigate the potential consequences for the size-structuring of species interactions. Using an extensive dataset spanning a large latitudinal range, we have shown that increasing temperature is associated with changes in body mass and dietary size-selectivity across Southern Ocean myctophids, a key component of pelagic food webs, resulting in predator communities that are closer in size to their prey. As a result, warming might alter prey population dynamics and reduce top-down control, potentially reducing community stability. The shift in predator-prey size relationships could also drive a reduction in the diversity of predator-prey interactions and a loss of redundancy within ecological networks, which may reduce their resistance to perturbations. The trends identified in this study provide a basis for mechanistic models to investigate the potential consequences of warming scenarios for the structure of biotic interactions and the stability of ecosystems. Efforts to investigate these relationships in other regions and for other taxa will aid the search for macroecological patterns that can be used to predict ecosystem responses to climate change.

## Methods
All data used in this study were collected following standard protocols and ethical approval from the British Antarctic Survey and the Environmental Protocol (1991) of the Antarctic Treaty.

### Fish sampling
Myctophids were collected during three research surveys conducted in austral spring (JR161, Oct-Dec 2006), summer (JR177, Jan-Feb 2008) and autumn (JR200, Mar-Apr 2009) in the Scotia Sea in the Atlantic sector of the Southern Ocean. Fish were sampled at stations across a transect spanning the entire Scotia Sea, from the Antarctic Polar Front to the sea ice zone. The exact location of these stations varied between cruises but was similar across years, with a broad latitudinal range sampled during each cruise (Figs. S5–S6). Sampling was conducted using a depth-stratified 25 m$^2$ rectangular mid-water trawl net (RMT25), deployed at depth ranges of 0–200, 200–400, 400–700, and 700–1000 m (Fig. 1). The nets had a cod end mesh size of 5 mm. Hauls were conducted during both light and dark conditions in spring and summer, but only darkness during autumn, due to a reduced daylight period.

Fish were processed on-board and identified to species level where possible, with standard length (SL) measured to the nearest millimetre. A random subsample of 25 fish per species (or all individuals in the case of small catches) were set aside for stomach dissection. These stomach samples were then frozen at −20 °C for later laboratory analysis, where the stomach contents were thawed and identified to the lowest taxonomic level possible. For each stomach, the number of individuals and average weight of each prey taxon was recorded using a motion compensated balance. The resulting datasets can be accessed via the UK Polar Data Centre[53,54].

For this study, fish SL was converted to mass in grams using species-specific length-weight equations from the British Antarctic Survey's long-term records (supplementary Table S25) for those individuals which did not have empty stomachs. This was done for ten species (see supplementary Table S25), while data for a further two species were omitted due to very low sample sizes ($n = 7$ for *Gymnoscopelus opisthopterus*, $n = 1$ for *G. piabilis*). The final datasets used in this study consisted of 3707 prey records from 1576 fish stomachs (Table S26), in addition to a larger set of fish body size estimates from 6143 individuals (the majority without stomach content data; Table S27) and species-specific abundance estimates for each sampling location.

### Zooplankton sampling
Macrozooplankton samples were collected from RMT25 nets, while mesozooplankton were sampled using paired Bongo nets (mesh size 50 μm), which were deployed to a depth of 400 m during daylight hours[55–57]. Zooplankton samples were preserved in 4% formalin with seawater and analysed in the laboratory, with taxa identified to the lowest possible taxonomic resolution. The total wet weight (g) was calculated for each macrozooplankton taxon using a motion compensated balance and divided by the number of individuals to estimate the mean body mass for each taxon. Mesozooplankton taxa were assigned an average dry mass (DM, mg) from published sources, which were converted to wet mass (WM, g) using general DM to WM conversion factors in Atkinson et al.[58]. Abundance values for macro- and mesozooplankton (standardised to individuals m$^{-2}$) were calculated using the estimated area sampled by the nets. Copepods dominated the zooplankton community by abundance, constituting over 70% of total density on average across hauls, followed by polychaetes and chaetognaths and, to a lesser extent, pteropods and ostracods (Table S28). The original zooplankton data are as presented in[57] and can be accessed from the UK Polar Data Center[59].

### Environmental covariates
We extracted daily sea-surface temperature (SST) values for the coordinates of each station from the 1/12° gridded Copernicus Global Ocean Physics Reanalysis product GLORYS12V1[60]. To investigate the consistency of results at depth, we also extracted modelled temperature data from the GLORYS12V1 -1062 m depth bin, which is the closest match to the lower depth limit of the trawls. Temperature data were averaged for the 30 days prior to and including the day of sampling. To identify the potential influence of local productivity on myctophid feeding relationships, we also extracted surface chlorophyll-a (Chl-a) values from the Copernicus-GlobColour dataset, which has a spatial resolution of 4 × 4 km[61]. As with the temperature data, daily Chl-a values at each station were averaged for the 30 days prior to and including the day of sampling. See Figure S7 for an overview of the relationship between temperature and latitude. The remaining methods refer to analyses involving SST but see Supplementary Information for an overview of the results of modelling with temperature at depth. We did not consider the effects of spatial heterogeneity in fishing effort as there is currently no targeted myctophid fishery in the Southern Ocean. Fish constitute the majority of bycatch by the winter krill fishery in the Scotia Sea but appear to consist predominantly of members of the Channichthyidae and Nototheniidae[62]. Overall annual average bycatch weights across all bycatch taxa (0.1-51.3 tonnes) are low compared to the estimated biomass of mesopelagic fish in the Scotia Sea (~4.5 million tonnes) and would therefore be expected to have negligible impact on community structure[62].

### Statistical analyses
Linear mixed models (LMMs) were used to investigate the relationship between the environmental variables and multiple metrics related to myctophids and their prey, using the predator-prey body size dataset. PPMR was calculated as the body mass of each fish predator (g) divided by the abundance-weighted average prey mass (g) in its stomach. LMMs were fitted using the function 'lme' in the package 'nlme'[63] with either PPMR, predator body mass, or abundance-weighted mean prey body mass as response variables (each subject to log$_{10}$ transformation to meet the assumptions of normality, homogeneity and independence of residuals). No strong collinearity was identified between SST and chl-a (Spearman's rho: −0.077, $p = 0.002$), therefore these were both entered as explanatory variables in the same model, including their interaction term. Model selection was then conducted to identify the best specification of fixed effects (SST and Chl-a) and random effects (nesting the variables 'year' and 'predator species'). The use of weighted variance structures to account for heterogeneity in residual

variance by year or species was also investigated during model selection. The absence of spatial autocorrelation in model residuals was confirmed using Moran's I (Table S7), therefore autocorrelation structures were not included in the models. The best model was determined by AIC comparison and visual diagnostics (heteroscedasticity and normality of residuals). All models incorporated a combined constant variance structure to account for heteroscedasticity in the errors within both year and predator species. The final selected models all included a random intercept for year and a random slope for SST by predator species. Chl-a had no significant main or interactive effects on the response variables and was therefore omitted from further analyses. See Tables S1–S7 for an overview of the model selection process and Moran's I results for these models.

The selectivity of predators for different prey sizes was estimated by fitting kernel density distributions to the prey body masses identified in predator stomachs (realised distribution) and to the comparable range of prey body masses sampled from the environment (environmental distribution)[64]. The environmental distribution represents the expected predator diet if feeding is based solely on density-dependent foraging, while the realised distribution generally represents the combination of such neutral processes and the active selection for specific prey sizes[64]. This approach assumes that the diets of these predators are generalist and primarily size-constrained, which is supported by previous studies of Southern Ocean myctophid diets[32,33]. Using the ratio of the realised and environmental distributions, a preference distribution can be calculated, representing the selectivity of predators for different prey sizes. To link the predator diets to the distribution of potential prey sizes in the environment, we grouped predators and zooplankton samples which were collected in the same area and within a few days of one-another, resulting in a total of 24 separate sampling locations spanning the study region. Within these groups, we then aggregated predators from the same species into size-classes of $10^{0.05}$ g to ensure that enough prey were present in the combined diets to reliably estimate a density distribution, whilst ensuring there were enough data points for later analysis ($n = 164$). The final size classes ranged from $10^{-0.525} = 0.30$ g to $10^{1.575} = 37.58$ g. For each aggregation, an average temperature was estimated from the constituent stations. We used the mean value of the preference distribution for each size class to represent the average preferred prey size of predators at each temperature. We then used a LMM to investigate the relationship between preferred prey size and the interaction between temperature and predator size-class, following the same approach to model specification and selection as described above. The final model included random intercepts for year and predator species, and a combined variance structure for year and predator species (see Tables S29-S30 for an overview of the model selection process).

To differentiate the potential individual-level and community-level mechanisms underlying trends in body size with temperature, we also conducted analyses of predator body size and community composition using a larger dataset of individual body sizes and species abundance estimates ($n = 6143$). We fitted a Generalised Least Squares (GLS) regression model of species diversity (Shannon–Wiener (log e) diversity index) as a function of SST and Chl-a to investigate whether there was any change in community structure with environmental conditions. For this analysis, densities of each species caught during each haul were estimated by multiplying counts by the product of the distance towed multiplied by the nominal net mouth area (25 m²), and then standardised to values of individuals per 1000 m⁻³. A square-root transformation was then applied to the density estimates to reduce the weighting of dominant species. An LMM was fitted to the relationship between body mass and the interaction between SST and Chl-a at the community level, before linear models of body size and SST were fitted for each predator species individually, to identify whether community-level trends in size with temperature were present at the population level. The optimal model structure for each species-level analysis

varied, and very low but statistically significant levels of spatial autocorrelation were identified for a small number of species and dealt with by incorporating spatial autocorrelation functions. See Tables S16–S24 for model selection of the optimal variance weighting, random and fixed effects structures, Moran's I test results and implemented autocorrelation structures and model outputs.

### Reporting summary

Further information on research design is available in the Nature Portfolio Reporting Summary linked to this article.

## Data availability

The processed data have been deposited in the Zenodo database[65]. The original raw fish stomach contents and zooplankton data used in this study are available in the UK Polar Data Centre[53,54,59] and an associated publication[57]. The SST and surface chlorophyll-a data used in this study are available in the Copernicus Marine Service database[60,61]. Source data are provided with this paper.

## Code availability

The code used to process raw data and conduct analyses are available in Zenodo[65].

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

## Acknowledgements
Thanks to the crew and scientists involved in the surveys during which the data used in this study were collected. Thanks to also Dr Michelle Taylor and Dr Anna Sturrock at the University of Essex and Dr Phil Hollyman at the British Antarctic Survey for providing advice regarding the analyses and reviewing drafts of the paper. This work was supported by the Natural Environment Research Council (NERC) and ARIES Doctoral Training Partnership [NE/S007334/1] (P.E.K.), the NERC-funded SeaDNA project [NE/N005996/1] (E.O.G.), the NERC-funded British Antarctic Survey Antarctic Logistics and Infrastructure National Capability programme CONSEC (S.H., G.T. and R.S.), the German Research Foundation [DFG-FZT 118, 202548816] (B.R.) and the EU COST action Sea-Unicorn [CA19107] (P.E.K.).

## Author contributions
Study conceived by P.E.K., E.O.G. and L.L. Data collated by G.T., R.S. and P.E.K. Analyses conducted by P.E.K., supported by B.R. for selectivity analysis. Manuscript written by P.E.K., with substantial contributions from E.O.G., L.L., G.T., R.S., B.R. and S.H.

## Competing interests
The authors declare no competing interests.
