## [Peer Review File · Nature Communications]

Temperature alters the predator-prey size relationships and size-selectivity of Southern Ocean fishREVIEWER COMMENTS

Reviewer #1 (Remarks to the Author):

I very much enjoyed reading this manuscript, and find myself in the rather unusual position of having very few comments. My only substantive concern, detailed below, is that lack of information provided about spatial autocorrelation. Such information should absolutely be included in a revised manuscript, and ideally included in your mixed models.

L41-67. Although I am personally a little sceptical of the mechanistic explanations proposed to explain observed size reductions, it is nonetheless quite clear that such declines are occurring. And so I very much enjoyed reading this introduction, which nicely links the side-dependence of size declines with the use of predator-prey mass ratios as indicators of trophic interactions, and then to ecosystem function. Nice work.

Figure 1. My first impression when viewing this figure is that there is likely to be significant spatial autocorrelation in the data, which has the potential to significantly undermine your conclusions. I was therefore very pleased to see that you considered spatial autocorrelation in the data (L322). However, I was then disappointed that I cannot find any information about these analyses in the main text or supplementary information to reassure the reader. I apologise if I have missed this somehow, but I think that it will be important to present this information in a revised manuscript (and ideally you would account for the presence of spatial autocorrelation in your mixed models).

Reviewer #2 (Remarks to the Author):

The study is carefully designed and the manuscript well written. I like how authors have combined predator and prey observations with the sampling of potential prey in the field. The subject is timely and impactful and helps to better understand the effects of temperature on community structure and biotic interactions. The outcomes demonstrate the effects of temperature after environmental and biotic filters have shaped the community. This, in my view, makes the work very informative for projecting the long-term effects of warming.

My primary concern is the emphasis on temperature effects on physiology and body size that authors have used to contextualize the paper. However, this paper examines the body size effects of temperature on the myctophid community. Hence, declines in body size can both be driven by changes in body sizes due to a physiological temperature effect on species, as well as community assembly processes that select differently sized species along the temperature gradient.

In my view, all observed outcomes can also be interpreted through the lens of community assembly, without necessarily considering potential physiological temperature effects on body size. For instance, there is a noticeable absence of zooplankton between -2 and 0 \log_{10} gram in the warmer waters (Fig 3B). This absence forces large fish to feed on relatively small prey (Fig 3A and indicated in lines 201-202). Consequently, the larger fish are outcompeted by smaller ones in warmer waters, which can be expected when both small and large fish target the same small size prey (as shown in the right part of Fig 3A). As a result, assembly processes favor small fish in warm waters and shift the size distribution of the myctophid community towards smaller body sizes (Fig 2B).

Importantly, this alternative narrative is not intended as a criticism of the current line of thinking but aims to encourage the authors to consider a different, non-physiological, explanation for the observed patterns. The above line of reasoning also introduces some uncertainties to two results of the manuscript:

- the decline in size preference in large fish (Fig 3A) could be driven by the absence of large prey in the environment, instead of, as stated in the manuscript, large predators preferring smaller prey in warm waters (lines 172-173),
- the temperature-size response of fish can solely be explained through a zooplankton community shift with temperature that, through assembly processes, selects for smaller sized myctophids. This contradicts the statement that there is a stronger temperature-size response among the myctophid predators than their smaller prey (line 118 and 155-156).

I would recommend that the authors leverage the community perspective and emphasize

the distinction between individual/species and community-level processes in their interpretation of the results. This may only need some re-writing of the manuscript.

Possibly, authors may be able to uncover the relative importance of the two processes (selection of different species by community assembly along the temperature gradient versus body size shifts with temperature in species). Authors should have information on the species composition in warm and cold waters. In addition, authors may be able to use the information from two co-authors (Saunders and Tarling, Bergmann's rule paper) to estimate the expected decline in body size with temperature within each species. However, this analysis is not essential for the paper.

--- minor comments ---

1) Initially, I was worried that there was too strong a focus on temperature without testing any other potential factors that may affect predator and prey size. It might be good to mention in the result section that the effects of CHL-A were insignificant (it is now only stated at the end of the paper). It might also be good to mention that fishing intensity is low (is it?) as fishing could have a much stronger effect on body size than temperature.

2) My suggestion would be to avoid stating that there is a stronger temperature-size response among the myctophid predators than their smaller prey and mainly discuss that this is a likely explanation of the observed patterns.

Authors make this statement already in line 118 and 155-156, while the available evidence for such a statement is limited at that stage. The observations of prey only consist of the selected prey that are in the stomach, and these do not reflect the size distribution of the "prey" populations in the field. The prey populations could still have a strong temperature-size response.

Only after seeing the dashed lines in Fig 3B (lines 168-176), there is some evidence that the temperature-size response is small on the prey (i.e. average size of prey is unchanged with temperature). Nonetheless, there still seems to be a clear decline in the 5 to 10% largest

zooplankton with temperature in Fig 3B.

3) Lines 261-268: were all stations sampled in all three surveys (and seasons/years)? I expect this is the case but please make it explicit. If all warm water communities were sampled in a specific season/year, this could generate a bias that might not be accounted for with the random effect.

4) The mean prey size in the stomach is in some sampling stations >0.1 gram (Fig 2C), whereas the body mass distributions of zooplankton in the environment tend to be smaller (Fig 3B). Did authors obtain samples where the prey size in the stomach was not observed in the environment? How did authors estimate size preference in that case?

*** Note that original reviewers' comments are in bold. Our responses are numbered sequentially, written in plain font, and contain quotations to the revised text in italics with line numbers corresponding to the revised version of the manuscript with all track changes accepted ***

Reviewer # 1:

I very much enjoyed reading this manuscript, and find myself in the rather unusual position of having very few comments. My only substantive concern, detailed below, is that lack of information provided about spatial autocorrelation. Such information should absolutely be included in a revised manuscript, and ideally included in your mixed models.

Response #1: Many thanks for your very positive feedback on our manuscript, and for the helpful comment regarding the need to provide more detail on spatial autocorrelation (this is addressed in response #3).

L41-67. Although I am personally a little sceptical of the mechanistic explanations proposed to explain observed size reductions, it is nonetheless quite clear that such declines are occurring. And so I very much enjoyed reading this introduction, which nicely links the side-dependence of size declines with the use of predator-prey mass ratios as indicators of trophic interactions, and then to ecosystem function. Nice work.

Response #2: Thank you. In response to Reviewer #2, we have broadened our discussion of the possible mechanisms underlying temperature-driven shifts in myctophid community size structure to include assembly processes in addition to our original focus on physiological mechanisms. We hope that this reduces any scepticism you felt regarding the original draft.

Figure 1. My first impression when viewing this figure is that there is likely to be significant spatial autocorrelation in the data, which has the potential to significantly undermine your conclusions. I was therefore very pleased to see that you considered spatial autocorrelation in the data (L322). However, I was then disappointed that I cannot find any information about these analyses in the main text or supplementary information to reassure the reader. I apologise if I have missed this somehow, but I think that it will be important to present this information in a revised manuscript (and ideally you would account for the presence of spatial autocorrelation in your mixed models).

Response #3: We have improved the clarity of the statement in the methods regarding the use of Moran's I test to make it clearer that we found no significant evidence of spatial autocorrelation in our predator-prey size models (Ln 370-372). We have also added tables to the supplementary information providing an overview of the Moran's I estimates and p-values for each of the selected model structures (Tables S7, S15, S17, S20, S23). We only find evidence of significant spatial autocorrelation in new species-level analyses of myctophid body mass which we performed to address a concern by Reviewer #2 (cf. Response #5), but we now provide the spatial autocorrelation structure implemented in all the relevant models (e.g. see Table S20 which we include below as an example of the new clarity on spatial autocorrelation analyses). Note that we have also updated all the statistical outputs in the manuscript with outputs from models fitted with 'method = REML' rather than our original use of 'method = ML', to match the recommendations of Zuur et al. (2009). Changes to resulting statistical outputs are extremely minor and there is no qualitative change to any of the results in the original manuscript.

Zuur, A.F., Ieno, E.N., Walker, N.J., Saveliev, A.A. and Smith, G.M.. Mixed effects models and extensions in ecology with R (Vol. 574, p. 574). New York: springer. (2009).

Ln 370-372: *The absence of spatial autocorrelation in model residuals was confirmed using Moran's I (Table S7), therefore autocorrelation structures were not included in the models.*

Table S20: Identification of the optimal fixed effects structure for the models involving predator mass for each myctophid species. *The table shows the fixed effects structures for the linear models of the relationship between predator size and sea-surface temperature (SST) for each species, using a larger dataset of myctophid body sizes (n = 6,143). Each model includes the optimal random effects and variance weighting structure identified in Table S19. The most parsimonious model based on Akaike's Information Criterion (AIC) and retaining only significant fixed effects is highlighted in bold. The results of a Moran's I test for spatial autocorrelation are provided for the optimal models, along with the optimal correlation structure implemented based on AIC for any models with significant autocorrelation.*

Species	Fixed effects structure	AIC	Moran's I	P-value	Autocorrelation structure
E. carlsbergi	SST	-1340.507			
	Null	-1342.335	<0.001	0.717	

E. antarctica	SST	1542.642	0.026	<0.001	Rational
	Null	1569.967			
G. fraseri	SST	34.019	0.060	0.004	Exponential
	Null	68.081			
G. nicholsi	SST	55.946	0.004	0.689	
	Null	71.849			
G. braueri	SST	1356.068	0.009	0.017	Exponential
	Null	1366.125			
K. anderssoni	SST	824.964	-0.010	0.034	Spherical
	Null	1001.208			
N. achirus	SST	-72.145			
	Null	-73.502	-0.104	0.091	
P. tenisoni	SST	-469.989	-0.004	0.950	
	Null	-442.164			
P. bolini	SST	-83.925	-0.013	0.094	
	Null	-76.457			
P. choriodon	SST	-93.411	-0.021	0.925	
	Null	-78.834			

Reviewer # 2:

The study is carefully designed and the manuscript well written. I like how authors have combined predator and prey observations with the sampling of potential prey in the field. The subject is timely and impactful and helps to better understand the effects of temperature on community structure and biotic interactions. The outcomes demonstrate the effects of temperature after environmental and biotic filters have shaped the community. This, in my view, makes the work very informative for projecting the long-term effects of warming.

Response #4: Thank you very much for your overall endorsement of our work.

My primary concern is the emphasis on temperature effects on physiology and body size that authors have used to contextualize the paper. However, this paper examines the body size effects of temperature on the myctophid community. Hence, declines in body

size can both be driven by changes in body sizes due to a physiological temperature effect on species, as well as community assembly processes that select differently sized species along the temperature gradient.

Response #5: We agree that it is important to recognize the potential for community-level processes to influence the observed relationships and to provide a more balanced appraisal of the physiological (population-level) and environmental filtering (community-level) mechanisms. To tackle this, we have revised the introduction to include an overview of how temperature-mediated community-assembly processes could result in declines in body size with temperature at the community level (Ln 43-45 & Ln 53-56).

We have also conducted new analyses to explore the possibility that community assembly processes are driving the observed changes in predator size with temperature (i.e. through changes in species composition). This involved the use of a larger dataset of myctophid body masses containing length measurements for 6,143 individuals of the ten species investigated here (only 1,576 of these were originally kept for stomach content analyses and that subset therefore underlies our core analyses for Figure 2 in the main text). This larger dataset includes a local density estimate of each species for every haul and is therefore useful for demonstrating an increase in myctophid community diversity with temperature, as larger species become less dominant in warmer regions (Figure S4; Tables S22-S24), supporting the possibility that community assembly processes are contributing to the results e.g. the higher temperatures and shift in prey size distributions favours smaller predators. We also confirm the previously reported decline in predator body mass at the myctophid community level (Figure 2b) with this larger dataset (Ln 128-130; Figure S2; Tables S16-S18), and demonstrate that seven of the ten myctophid species display a decline in size with increasing temperature (Ln 130-131; Figure S3; Tables S19-S21), which supports the possibility that physiological processes (i.e. population-level) are contributing to our results. We provide an overview of these new analyses in the methods section (Ln 405-423) and discuss how both community assembly processes and individual physiology may play a role in driving the observed trends in PPMR and predator body mass with temperature (Ln 172-186 & Ln 237-245).

Ln 43-45: *At the community level, warming may alter assembly processes through environmental filtering, competition, or trophic interactions, which may result in communities dominated by smaller-bodied species.*

Ln 53-56: *Furthermore, such altered prey size distributions may favour smaller-sized predator species, providing them with a competitive advantage and thereby shifting the fish community composition towards smaller body sizes.*

Ln 128-130: *A similar decline in predator body size was also found when using a larger dataset of fish body masses ($n = 6,143$, the majority without stomach content data; Figure S2, Tables S16-S18).*

Ln 130-131: *Seven of the ten myctophid species also displayed significant declines in size with increasing temperature (Figure S3, Tables S19-21).*

Ln 172-186: *There was a significant increase in Shannon diversity of the myctophid community with increasing SST, driven by a shift in species abundances from communities dominated by a few large-bodied species (e.g. *E. antarctica*) at cold high latitudes to a more even distribution of abundances in the more northerly warmer regions (Figure S4, Tables S22-24), as previously documented⁴⁰. This indicates that the link between temperature and body size at the community level may be driven in part by community assembly processes which select for species of different sizes as it becomes warmer, e.g. smaller predators are able to outcompete larger ones under the altered prey size distribution and relatively lower metabolic demands. However, our analyses of the relationship between body mass and SST at the population level also revealed significant declines in size with increasing temperature for many of the myctophid species, both for large-bodied taxa such as *E. antarctica* and for small species like *K. anderssoni* (Figure S3, Tables S19-21). The observed trends at the community level therefore are not explained by community assembly processes alone, but also by temperature effects on populations, likely mediated by physiological responses to warming.*

Ln 237-245: *These changes in size-selectivity may also explain the increasing prevalence of smaller myctophid species in warmer regions (Figure S4), as they can capitalise on the available prey field and outcompete their larger counterparts. The increasing dominance of smaller myctophids, which feed preferentially on larger prey in warmer regions, is likely to drive the observed decline in overall PPMR across the predator community. Thus, we suggest that the observed patterns in myctophid size and foraging with temperature are likely to be the result of a combination of interacting processes acting at both the population and community levels, and we encourage further efforts to disentangle them.*

Ln 405-423: *To differentiate the potential individual-level and community-level mechanisms underlying trends in body size with temperature, we also conducted analyses of predator body*

size and community composition using a larger dataset of individual body sizes and species abundance estimates ($n = 6,143$). We fitted a Generalised Least Squares (GLS) regression model of species diversity (Shannon–Wiener ($\log e$) diversity index) as a function of SST and Chl-a to investigate whether there was any change in community structure with environmental conditions. For this analysis, densities of each species caught during each haul were estimated by multiplying counts by the product of the distance towed multiplied by the nominal net mouth area (25 m^2), and then standardised to values of individuals per $1,000 \text{ m}^3$. A square-root transformation was then applied to the density estimates to reduce the weighting of dominant species. An LMM was fitted to the relationship between body mass and the interaction between SST and Chl-a at the community level, before linear models of body size and SST were fitted for each predator species individually, to identify whether community-level trends in size with temperature were present at the population level. The optimal model structure for each species-level analysis varied, and very low but statistically significant levels of spatial autocorrelation were identified for a small number of species and dealt with by incorporating spatial autocorrelation functions. See Tables S16-S24 for model selection of the optimal variance weighting, random and fixed effects structures, Moran's I test results and implemented autocorrelation structures, and model outputs.

In my view, all observed outcomes can also be interpreted through the lens of community assembly, without necessarily considering potential physiological temperature effects on body size. For instance, there is a noticeable absence of zooplankton between -2 and 0 \log_{10} gram in the warmer waters (Fig 3B). This absence forces large fish to feed on relatively small prey (Fig 3A and indicated in lines 201-202). Consequently, the larger fish are outcompeted by smaller ones in warmer waters, which can be expected when both small and large fish target the same small size prey (as shown in the right part of Fig 3A). As a result, assembly processes favor small fish in warm waters and shift the size distribution of the myctophid community towards smaller body sizes (Fig 2B).

Response #6: We agree that, as you suggest, the decline in zooplankton size distribution in warmer waters could indeed alter competitive dynamics between large and small predators, leading to a greater proportion of small-bodied species (though body sizes between -2 and 0 \log_{10} g are still present, if rare, in warmer regions – see Response #13). Our new analysis of myctophid community composition provides some support for this mechanism, though our

new species-level analyses suggest that physiological processes are also contributing, and we thus discuss both as likely mechanisms driving the observed trends (see Response #5).

Importantly, this alternative narrative is not intended as a criticism of the current line of thinking but aims to encourage the authors to consider a different, non-physiological, explanation for the observed patterns. The above line of reasoning also introduces some uncertainties to two results of the manuscript:

- **the decline in size preference in large fish (Fig 3A) could be driven by the absence of large prey in the environment, instead of, as stated in the manuscript, large predators preferring smaller prey in warm waters (lines 172-173)**

Response #7: We agree that the likely explanation for the decline in size preference for large predators is that larger prey become so scarce that the predators are forced to feed on smaller prey. This was the intended explanation in the original draft, but we acknowledge that our wording did not make this clear. We have adjusted our discussion of the changes in size selectivity to emphasise that large predators are forced to feed on smaller prey under warmer conditions (Ln 228-232).

Ln 228-232: Under warmer conditions, the shift in the distribution of suitable prey sizes towards intermediate body masses restricts the feeding behaviour of large predators and forces them to feed sub-optimally on smaller prey while small predators actively select for these abundant intermediate prey sizes, possibly because they provide greater per capita energy intake.

- **the temperature-size response of fish can solely be explained through a zooplankton community shift with temperature that, through assembly processes, selects for smaller sized myctophids. This contradicts the statement that there is a stronger temperature-size response among the myctophid predators than their smaller prey (line 118 and 155-156).**

Response #8: Our new analyses (Response #5) provide some support for assembly processes selecting for smaller myctophids but also suggest that physiological mechanisms are acting at the population-level to drive a decline in the body size of individuals within most species. Based on your minor comment 2, we have adjusted the text to make our statements about relative changes to predator and prey body sizes a description of the likely driver of the observed trends (see Response #11).

I would recommend that the authors leverage the community perspective and emphasize the distinction between individual/species and community-level processes in their interpretation of the results. This may only need some re-writing of the manuscript. Possibly, authors may be able to uncover the relative importance of the two processes (selection of different species by community assembly along the temperature gradient versus body size shifts with temperature in species). Authors should have information on the species composition in warm and cold waters. In addition, authors may be able to use the information from two co-authors (Saunders and Tarling, Bergmann's rule paper) to estimate the expected decline in body size with temperature within each species. However, this analysis is not essential for the paper.

Response #9: Thanks for this suggestion, which guided the new analyses (see Response #5) and helped to strengthen the manuscript.

Minor comments

1). Initially, I was worried that there was too strong a focus on temperature without testing any other potential factors that may affect predator and prey size. It might be good to mention in the result section that the effects of CHL-A were insignificant (it is now only stated at the end of the paper). It might also be good to mention that fishing intensity is low (is it?) as fishing could have a much stronger effect on body size than temperature.

Response #10: We have added a statement to the results to clarify that Chl-a was not a significant predictor in our models (Ln 124-126). We have also provided an explanation in the methods section for the omission of fishing effort as a predictor in our models (Ln 349-356).

Ln 124-126: *Chlorophyll a was initially used as a further explanatory variable but it was not significant in any model and was therefore excluded during model selection (Tables S1-S7).*

Ln 349-356: *We did not consider the effects of spatial heterogeneity in fishing effort as there is currently no targeted myctophid fishery in the Southern Ocean. Fish constitute the majority of bycatch by the winter krill fishery in the Scotia Sea but appear to consist predominantly of members of the Channichthyidae and Nototheniidae. Overall annual average bycatch weights across all bycatch taxa (0.1-51.3 tonnes) are low compared to the estimated biomass of*

mesopelagic fish in the Scotia Sea (~4.5 million tonnes) and would therefore be expected to have negligible impact on community structure.

2). My suggestion would be to avoid stating that there is a stronger temperature-size response among the myctophid predators than their smaller prey and mainly discuss that this is a likely explanation of the observed patterns.

Response #11: We have adjusted statements to put forward the idea of a greater decline in predator size relative to prey as a potential explanation for the observed trends in PPMR, rather than a statement of fact (Ln 131-133). We have also removed the statement that greater oxygen limitation in larger than smaller organisms drives a stronger temperature-size response in the predator community than their prey (Ln 153-156 of the original manuscript), given your comments and our new discussion.

Ln 131-133: Together, these results suggest that the decline in PPMR is driven by a greater decrease in the size of these predators relative to their prey as temperature increases.

3) Lines 261-268: were all stations sampled in all three surveys (and seasons/years)? I expect this is the case but please make it explicit. If all warm water communities were sampled in a specific season/year, this could generate a bias that might not be accounted for with the random effect.

Response #12: The majority of cruise stations were fixed to allow inter-cruise comparisons, although some were allowed to vary between cruises to allow adequate sampling of dynamic environmental conditions. Overall, the distribution of sampling locations remained quite similar and a broad latitudinal range of stations was sampled during each cruise. We have added two new figures to the supplementary material (Figures S5-S6, reproduced below) which depict the distribution of fish and zooplankton sampling stations during each cruise. We have also added a statement to the methods (Ln 299-301) identifying the variability in location of some stations.

Ln 299-301: The exact location of these stations varied between cruises but was similar across years, with a broad latitudinal range sampled during each cruise (Figures S5-S6).

Figure S5: Distribution of myctophid sampling stations from each cruise.

Figure S6: Distribution of zooplankton sampling stations from each cruise.

4) The mean prey size in the stomach is in some sampling stations >0.1 gram (Fig 2C), whereas the body mass distributions of zooplankton in the environment tend to be smaller (Fig 3B). Did authors obtain samples where the prey size in the stomach was not observed in the environment? How did authors estimate size preference in that case?

Response #13: These larger prey body sizes are in fact present at all temperatures, however the extremely high relative abundance of smaller organisms prevents these larger prey from being visible in the fitted probability density functions of Figure 3b. We have added a statement to the figure legend noting this issue (Ln 217-219). The maximum prey size in the environment ranged from -0.07 to -0.11 (log₁₀ g) in the coldest and warmest stations, respectively, which matches the maximum prey sizes in Figure 2c. Below, we provide a version of Figure 3b with

tick marks added along the upper portion of each plot, representing the presence of a given prey body size. We are reluctant to use this in the final manuscript as we feel it is cluttered and overly complex, but we hope this provides reassurance regarding the existence of larger prey sizes in the environment.

Ln 217-219: Note that in panel b, the large prey sizes (above approx. $-2 \log_{10} \text{g}$) are present at all temperatures, but their extremely low abundance relative to smaller organisms prevents them from being visible in the figure.

REVIEWERS' COMMENTS

Reviewer #1 (Remarks to the Author):

I am entirely satisfied by the authors' responses to my comments, and am happy to say that I have no further suggestions for revision. Congratulations on a fine paper!

Reviewer #2 (Remarks to the Author):

Thank you for addressing my comments and those of the other reviewer. The new analyses on species and community responses to temperature are nicely integrated in the manuscript. The present version of the manuscript is enjoyable to read, clear and timely.